# Comparison of Electrical Contacting Techniques to Carbon Fiber Reinforced Plastics for Self-Strain-Sensing Applications

Patrick Scholle *, Sören Rüther and Michael Sinapius

Institute of Mechanics and Adaptronics, Technische Universtität Braunschweig, 38106 Braunschweig, Germany; s.ruether@tu-braunschweig.de (S.R.); m.sinapius@tu-braunschweig.de (M.S.)
* Correspondence: p.scholle@tu-braunschweig.de

**Abstract:** The electrical conductivity of carbon fibers can be used to enable the design of intrinsically smart carbon fiber reinforced plastics (CFRPs). Resistance and impedance measurements of the structural material itself can then be used to measure physical stimuli such as strain or damage without requiring a dedicated sensor to be installed. Measuring the resistance with high precision requires good electrical contact between the measurement equipment and the conductive carbon fibers. In the literature, many different combinations of surface contacting material and surface preparation procedures are used, but only seldomly compared to one another. This article aims to compare frequently used electrical contact methods by analyzing their contact resistance to a pultruded CFRP rod. Furthermore, this study explores the change of contact resistance with increasing mechanical strain. The results show that contact resistance is highly dependent on both the material used for contacting the fibers as well as the surface preparation technique. From the combinations analyzed in this article, the electrodeposition in combination with a surface treatment using concentrated sulphuric acid shows the most promising results.

**Keywords:** self sensing; electrical contacting; carbon fiber; electrical resistance

## 1. Introduction

The measurement of the electrical resistance of carbon fiber reinforced plastics can be used to generate an intrinsically smart material and allows to quantify various properties such as strain[1], temperature[2] or damage[3]. This approach is also referred to as Self-Sensing, because it allows measuring external stimuli by only observing a property of the structural material itself—no additional sensor is required.

The electrical contact between the measurement equipment and structural part is a critical component of a Self-Sensing setup. Various methods to create the electrical connection have been developed and tested in the past. We present an overview of previously used methods for Self-Strain-Sensing in [4]. Generally speaking, electrical contact is made by connecting an electrical wire to the CFRP surface with some highly conductive material. Many authors choose to use a type of adhesive filled with conductive particles, thus forming an electrically conductive adhesive. Examples for this are silver filled epoxys (e.g., used in [5]), silver paint (e.g., used in [6]) or graphite cement (e.g., used in [5]). Other methods rely on the deposition of metals onto the CFRP surface, for example through sputtering (e.g., used in [5]) or electrodeposition (e.g., used in [7]).

In order to achieve electrical contact to the conductive fiber network of a CFRP, the carbon fibers have to be exposed on the surface of the material. Excessive resin rich surface zones occurring frequently in CFRP manufacturing processes electrically isolate the fibers and therefore have to be removed. Different methods to achieve this have been proposed in the past. Many authors [8,9] use abrasive paper of various grid sizes to clean the surface of the CFRP, which not only abrades matrix material but also the fibers themselves[10,11]. Other authors use concentrated sulphuric acid [7] or laser ablation [12] to prepare the surface while minimizing fiber damage.

In summary, many different contacting methods and materials are proposed and applied in the literature. Some research work exists that compares the Self-Strain-Sensing properties of different contacting methods. For example, Angelidis et al. [13] found vastly different gauge factors for similar CFRP materials depending on whether silver epoxy, silver paint, or carbon cement is used as contacting material. In the case of carbon cement, they attribute this to be caused by an insufficient bonding between the contact material and the fibers that result in delamination in the contact. Todoroki et al. [14] analyze the influence of surface preparation to Self-Strain-Sensing properties. The authors analyze two types of specimen, one where the surface is thoroughly polished with abrasive paper and another where this step is not performed. Using silver paint to establish electrical contact, the authors find vastly different gauge factors for both specimen types that they attribute to an unreliable electrical contact without surface polishing.

Other research work analyzes specific materials in more detail. For example, Todoroki et al. [7] analyze the durability of specially prepared electroplated copper contacts on CFRP sheets. The authors show, that the contact resistance of this type of electrical connection remains stable for up to $10^6$ strain cycles with a force amplitude equal to 20 % of tensile strength. Other contact methods however have not yet been assessed in such detail. Nevertheless, different methods are still frequently used in the literature to the present day.

Overall, there seems not to be a clear consensus on which method is preferable. In contrast, the widely accepted hypothesis appears to be that many different materials such as silver adhesive[15,16], silver coatings [17] or silver paint [18] are all equally applicable as an electrical connection. In our view, this hypothesis has not yet been assessed to a sufficient extent.

This article is aimed to thoroughly compare different contacting methods and materials for Self-Strain-Sensing applications. We describe and discuss experimental results that measure the contact resistance both without mechanical load and with mechanical load to the specimen. In the next section, we would like to first present some arguments that explain why it is necessary to discuss contact resistance in more detail.

## 2. Motivation: On the Influence of Contact Resistance on the Acquired Resistance in Four-Wire Measurements

The majority of experimental research performed on Self-Strain-Sensing CFRP uses a four-wire technique to measure the sample resistance. This is because—generally speaking—the four-wire technique allows to exclude cable and contact resistance from the measurement. In contrast, in a two-wire experiment, both cable and contact resistance are placed in series to the measured CFRP resistance and are therefore part of the measured signal. This is especially problematic when the specimen resistance is small because cable and contact resistance in that case can be a significant fraction of the total measured resistance. This is further complicated in Self-Strain-Sensing applications because the contact resistance might change when strain is applied. One could argue, that these problems can be avoided simply by using a four-wire measurement. However, is this assumption correct under all circumstances? This section will discuss this question based on a simple experiment and a very simplified modelling approach.

### 2.1. Influence of the Contact Resistance on a Digital Multimeter

The contact resistance can in many cases be excluded from a measurement by using a four-wire resistance measurement. This assumption is valid for small contact resistances, but has to be critically analyzed for very large contact resistances. In this article, resistances are measured with a Keithley DMM6500. In a simple experiment, the influence of additional wire or contact resistance is tested using a decade resistance. For this experiment, decade resistances are installed in series to the measurement wires of a four-wire measurement. A low resistance metallic wire with a resistance of $14\,\mathrm{m\Omega}$ is measured in this experiment. In one experiment, the resistance in series with the current contacts are successively increased. In a second experiment, the resistance in series with the voltage contacts are increased in the same manner.

The results of both experiments are displayed in Figure 1. Figure 1 (top) shows the relative difference between the resistance measured with an added series resistance $R_c$ and the base value without added resistance $R(0\,\Omega)$. The mean resistance obtained in a time window of approximately 10 s is used for this. The measurement results show that the influence of contact resistance is larger when a series resistance is added to the current contacts. For the current contacts, a resistance change of approximately 0.3 % can be observed for an additional contact resistance of 10 $\Omega$. This is plausible because a constant current is injected through the current contacts which requires more power if the contact resistance is large, which could result in problems for the circuit responsible for providing the constant current. For the voltage contacts, the same deviation is only observed for added contact resistances of more than 10 k$\Omega$. Similar observations can be made by using the standard deviation of the measurement signal in the time window as a method to quantify noise. Figure 1 (bottom) shows the results of this analysis. Again, the noise increases more significantly when contact resistance is added to the current contacts.

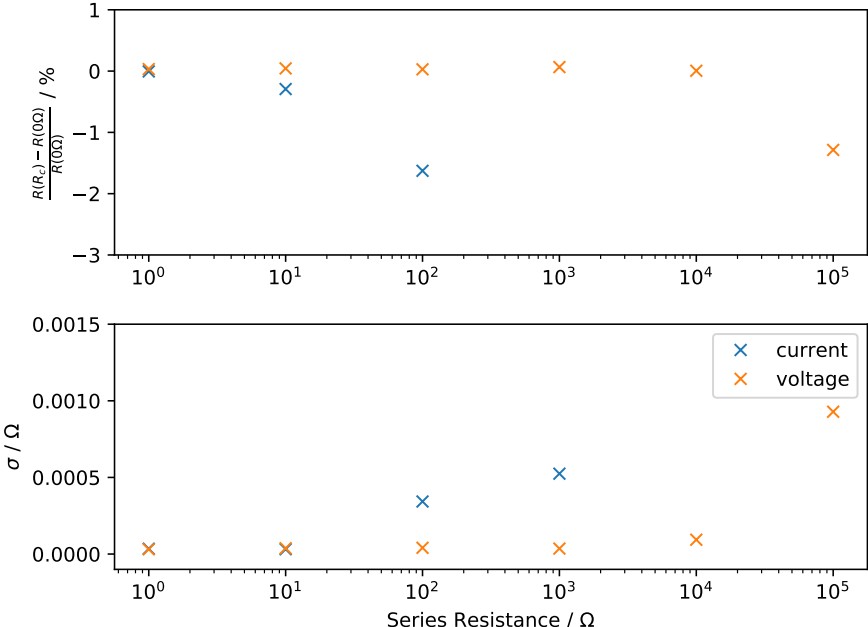

**Figure 1.** Influence of series resistance added to either current or voltage contacts of the Keithley DMM6500 for the four-wire measurement of a 14 m$\Omega$ resistor. (**top**): Relative change of measured resistance (**bottom**): Standard deviation of the acquired resistances in 10 s.

We would expect the same phenomena to occur if the contact resistance of the electrical connection changes during its life cycle or due to mechanical strain. The results, therefore, exemplify the necessity to look at contact resistance, especially in the case of contacts that are used to supply current.

Thus, a large change of contact resistance changes the acquired resistance in a four-wire measurement. This is problematic because we cannot know why the observed resistance changed after the measurement is done. It might be possible to correct for this influence by also measuring the two-wire resistances of all contact pairs and correcting for the influence after the measurement. Furthermore, it is possible that other equipment is more suited to work in this type of environment. There is however another reason to be interested in contact resistances in the case of measurements on CFRP surfaces. Due to the electrical anisotropy of the material, the electrical potential can be inhomogeneously distributed on the surface covered by an electrode. The influence of this is discussed in the following section based on a maximally reduced model.

### 2.2. Multi-Fiber Strain Sensors: A Maximally Reduced Model

Contact resistance has to be further analyzed critically in the case of highly anisotropic materials such as CFRP. The reason for this can be explained using a maximally simplified model of a two-fiber sensor displayed in Figure 2. A LTspice model https://www.analog.com/en/design-center/design-tools-and-calculators/ltspice-simulator.html (accessed on 21 September 2021) that demonstrates the following discussion is attached as a digital supplementary file to this article.

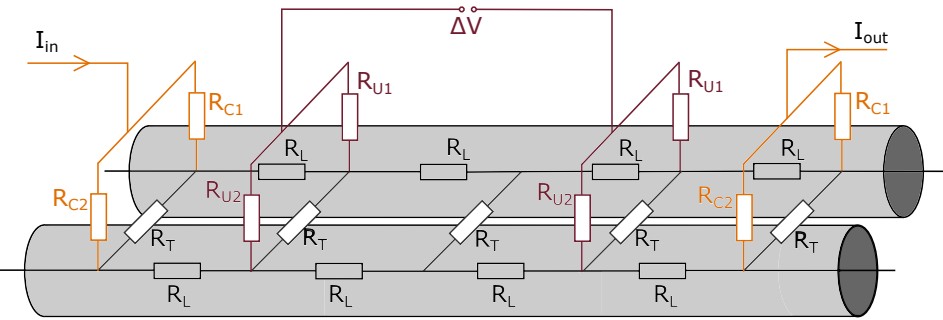

**Figure 2.** Schematic of the model of a hypothetical two-fiber-sensor.

Consider a strain sensor made from two parallel and identical carbon fibers represented by the series resistances $R_L$. Both fibers are connected in some finite spots with transverse resistances $R_T$ that are used here to represent fiber contact points. A four-wire measurement is used to measure the resistance of this hypothetical two-fiber sensor. Highly conductive cables with negligible resistance are used to connect the measurement equipment to the two-fiber-sensor. However, a contact resistance exists between the connecting cable and the two-fiber-sensor. The contact resistances on the current contacts are represented by $R_{C1}$ and $R_{C2}$ and the contact resistances on the voltage contact are represented by $R_{U1}$ and $R_{U2}$. The contact resistances therefore can be different in between the fibers, which is reasonable to assume in practical applications due to differences that can occur during the electrode application processes. All contact resistances are placed symmetrically for the sake of simplicity. This model obviously reduces the physical reality substantially, but it still allows to discuss under what circumstances contact resistances can be neglected. Surely, if we want to conclude that contact resistance can be fully compensated in four-wire experiments, the measured voltage $V$ ought to be fully independent of these resistances.

First, consider the simplest case where all contact resistances are of the same size. The current from the measurement equipment is then transported by both fibers equally. Assuming that those effects described in Section 2.1 can be neglected, no heating effects are observed and all resistors behave according to Ohms law, the measured voltage stays the same if we change the contact resistances $R_{U1}$ and $R_{U2}$ independently from one another. The measured voltage is therefore truly independent from the contact resistance. Let us name this baseline voltage as $V_{specimen}$.

What would happen if we increase one of the resistors on the current contact significantly? Obviously, current would initially only flow through one of the fibers. Depending on the ratio of the size of longitudinal and transverse resistors (which represents an electrical anisotropy in this analogy), current slowly starts to transfer to the second fiber. At the voltage measurement contact, fiber 1 and 2 now have two different potentials at the point of contact, one larger and one smaller than $V_{specimen}$. As long as the contact resistances of the voltage contacts are identical, the measured voltage drop is still the same $V_{specimen}$ measured before. This is because the voltage contact connects both fibers and the measured voltage is equal to a weighted mean of both electrical potentials. If the two contact resistances at the voltage contact are however different from one another, the measured voltage can be either larger or smaller than $V_{specimen}$, depending on the ratio $\frac{R_{U1}}{R_{U2}}$. The measured

voltage $\Delta V$ is thus dependent on the contact resistance distribution whenever the voltage terminals connect fibers with different electrical potentials.

What relevance does this simplified model have for practical applications? Generally speaking, contact resistances are likely to vary from one fiber to the next and depend on the manufacturing conditions. Furthermore, not all fibers are identical and neither are they oriented perfectly straight. Finally, due to the large electrical anisotropy of the material, any existing current inhomogeneity in between fibers does not easily equalize. This means, that the same dependency on contact resistance distribution observed in this simplified model should also exist in practical application. The contact resistance is therefore not fully compensated, even in four-wire measurements. The distribution of contact resistance over the contact area indeed has an influence on the acquired voltage. Furthermore, if the contact resistance distribution is changed due to external stimuli such as strain, this also influences the acquired voltage change. We conclude three main requirements that should be demanded for electrical contacts from the observations presented so far:

1. It is important to maximize the current homogeneity on the surface of a Self-Strain-Sensing CFRP.
2. The contact resistance should be minimized in order to allow a Digital Multimeter (DMM) to function correctly. It also seems reasonable to assume that the current homogeneity is improved when the contact resistance is minimized.
3. The influence of strain on the contact resistance distribution should be minimized.

We are not aware of a widely used measurement technique that allows quantifying the first requirement formulated here, but developed a viable solution to this research gap in [19] that shows that current is indeed inhomogeneously distributed on a CFRP surface for some contacting methods. This article discusses an experimental study that further analyzes the second and third requirements. First, we quantify contact resistances for various contacting techniques. This allows us to discuss the second requirement and identify contacting techniques that can be used in the measurement of electrical potential in CFRP. Next, we analyze the change of acquired resistance due to strain as well as the progression of resistance over 200 strain cycles. It seems reasonable to assume that a stable contact resistance also indicates that the contact resistance distribution remains stable. This allows us to further identify those contacting techniques that are most promising for Self-Strain-Sensing applications.

## 3. Methods and Materials

In the case of a carbon fiber reinforced plastic, the formation of the electrical contact is impeded by the complex surface topology consisting of conductive carbon fibers and an insulating matrix. Before generating an electrical contact, the carbon fibers have to be made accessible by sufficiently removing any resin rich surface layer that acts as an electrical insulator. Methods for electrical contacting can therefore differ by their surface preparation technique as well as the material and technique used to generate an electrical contact and connect the measurement cable to the CFRP.

For all measurements, pultruded CFRP rods supplied by dpp pultrusion https://www.dpp-pultrusion.com/ (accessed on 27 September 2021) with a fiber volume fraction of approximately 60 % are used. The rods are made using T700SC carbon fibers ($d \approx 7\,\mu\text{m}$, $E = 230\,\text{GPa}$) and an epoxy matrix. Unlike many other CFRP materials, these rods do not have a very thick resin rich surface.

### 3.1. Surface Preparation Techniques of CFRP

A commonly used surface preparation technique is sanding with sanding paper of different grid sizes. As displayed schematically in Figure 3, the sanding process abrades both matrix material as well as fibers. It has been previously pointed out that surface polishing damages the fibers [11] and results in fiber damage accumulating earlier than in non sanded surfaces [10]. Other researchers [1,20] use a chemical etching process with concentrated sulphuric acid in addition or instead of the mechanical sanding step. Since

carbon fibers are not easily affected by sulphuric acid, it is possible that an etching process is better suited to remove surface resin without damaging the fibers. Both of these techniques are compared in this article.

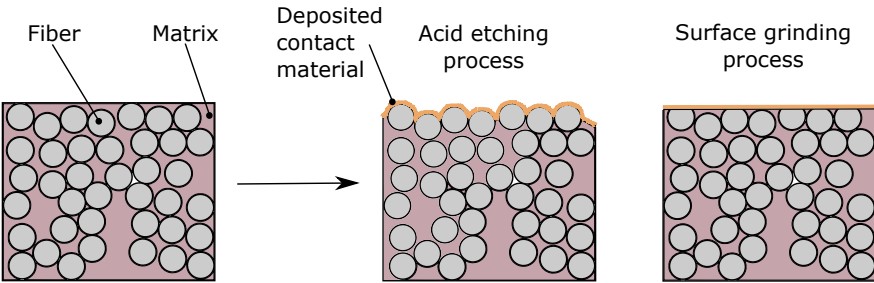

**Figure 3.** Schematic idea of the difference between mechanical sanding and acid etching surface treatments.

Sanded surfaces are prepared by manually sanding each contact for approximately 10 s with a #600 grid paper and light pressure. Etched surfaces are prepared by coating the surface with 96 % sulphuric acid for 1 min. The acid is then washed off under running water. All surfaces are cleaned with isopropanol before the contact material is placed onto the surface.

### 3.2. Surface Contacting Methods Based on Filled Adhesives

Table 1 summarizes the contacting materials based on conductive adhesives used in this article. Two different types of silver paint, one carbon paint and one silver filled epoxy are analyzed in this section. All coatings are brushed onto the CFRP after the specimen surface is prepared. The curing conditions for all materials are adapted from the datasheet and documented in Table 1. An electrical wire is placed onto the specimen surface and encapsulated with the conductive adhesive.

**Table 1.** Overview of contacting materials based on filled adhesives used in this article.

| General Term | Manufacturer | Description | Curing Conditions | Viscosity |
|---|---|---|---|---|
| Silver paint | Busch GmbH | Busch 5900 | dry at RT | - |
| Silver paint RS | RS | RS Pro Conductive Paint | dry at RT + cure at 130 °C for 10 min | 70 mPa s [21] |
| Silver epoxy | Panacol | Elecolit 3661 | cure at 150 °C for 15 min | 20,000 mPa s |
| Carbon paint | MG Chemicals | 838AR-Liquid | cure at 65 °C for 30 min | 154 mPa s |

### 3.3. Surface Contacting Methods Based on Electrodeposition

Another commonly used contacting technique is the electrodeposition of various metals onto the conductive fibers. Table 2 summarizes the different contacting materials based on electrodeposition used in this article. Two types of plating materials are tested. Electroplating is performed in a plating bath according to the manufacturer's recommendations. First adhesive tape is wrapped around the specimen at all positions that have to be submerged into the electrolyte but should not get any metal deposited. Next, the surface of the specimen is prepared as described in Section 3.1. The specimen is then clamped on one end between two copper sheets for electrical contact during electrodeposition. The other side of the specimen is then submerged into the electrolyte. Next, a current that results in a current density as described in Table 2 is set in a constant current source and the specimen is electroplated for the specified time. Due to the complex surface topology of CFRP with fibers exposed on the surface, it is difficult to precisely quantify the surface area. For simplicity, the surface area is calculated using the macroscopic dimensions of all surfaces where metal is deposited in this article. The same electroplating procedure is then

repeated on the other side of the specimen. Finally, if not stated otherwise, electrical contact to the deposited metal is generated by soldering a wire to it with a standard soldering iron.

**Table 2.** Overview of contacting materials tested for the electrodeposition.

| General Term | Manufacturer | Description | Current Density | Plating Time |
|---|---|---|---|---|
| Nickel electroplating | Tifoo | Nickel plating solution | $1\,\mathrm{A\,dm}^{-2}$ | 20 min |
| Silver electroplating | Tifoo | Silver plating solution | $0.4\,\mathrm{A\,dm}^{-2}$ | 20 min |

### 3.4. Estimating the Contact Resistance through Linear Approximation

A simple and effective way to estimate the contact resistance is described in [7]. The authors argue that the resistance between two points of a thin CFRP strip with sufficiently large electrodes and small cross section is linearly dependent on the distance between the contacts. They further argue that, without any contact resistance, a linear regression of the resistance over contact distance should intercept the y-axis at $0\,\Omega$. Consequently, any existing resistance at the intercept can be interpreted as the mean contact and cable resistance of all contacts. Furthermore, the repeatability of the contact resistance can be described by the coefficient of determination $R^2$, where a value close to 1 indicates a perfectly repeatable contact resistance for all contacts.

Figure 4 shows the dimensions of the pultruded specimen used in this part of the study. For each contacting method, one pultruded rod is equipped with six contacts on the lateral surface. All six of these lateral contacts are connected in two-wire configuration to a Keithley DMM6500. The cable resistance is measured and subtracted from the acquired resistance. All possible combinations of connections are measured for approximately 10 s. No large changes in measured resistance is observed in this time window. The mean resistance acquired for each contact pair in these 10 s is therefore used to describe the total specimen resistance.

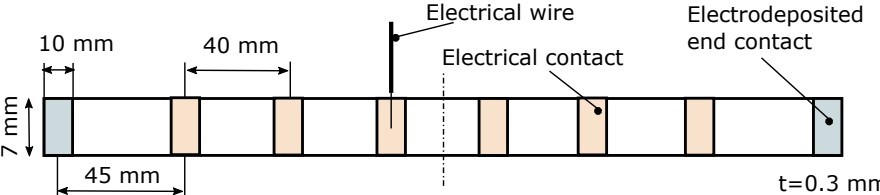

**Figure 4.** Specimen with a length of 300 mm used for the measurement of contact resistance to pultruded CFRP rods. The electroplated end contacts are not used for the discussion reported in this article.

### 3.5. Cross Section Studies

Throughout this study, microscopic cross section analyses are performed to assess the interface between electrical contact and CFRP in detail. For this, specimens with similar contact preparation techniques are manufactured separately and cut at the cross section of interest with a rotary cutter. These smaller specimens are then embedded in acrylic (Struers ClaroKit) and polished with a typical polishing cycle using an automated polishing system (Struers Tegramin-30). The specimens are then analyzed using general purpose microscopes.

### 3.6. Observing the Electrical Contact Resistance during Strain Cycles

The change of electrical contact resistance of various contacting methods due to strain is analyzed using two-wire measurements. Pultruded rods with a cross section of approximately $(8 \times 0.8)\,\mathrm{mm}^2$ are used. GFRP tabs are glued to the specimen in order to clamp the specimen into a universal testing machine. Figure 5a) shows the specimen dimensions used in this study. Figure 5c) shows part of the strain cycle used for all specimens, where the specimen is first strained and kept under strain for some seconds.

Afterwards, the specimen is repeatedly tensioned and released. All CFRP rods are strained with a universal testing machine with 200 cycles and a maximum strain of 0.3 %. A total of six different specimens are analyzed in these dynamic experiments. A total of five electrodes are painted or electroplated onto the lateral surface of the CFRP part. All electrode combinations are connected to a SCAN2000 multiplexer circuit and fed into a Keithley DMM6500 to allow for a quasi-simultaneous measurement of all contact pairs. The change of two-wire resistance of all contact pairs is used to estimate the change of contact resistance throughout the measurement.

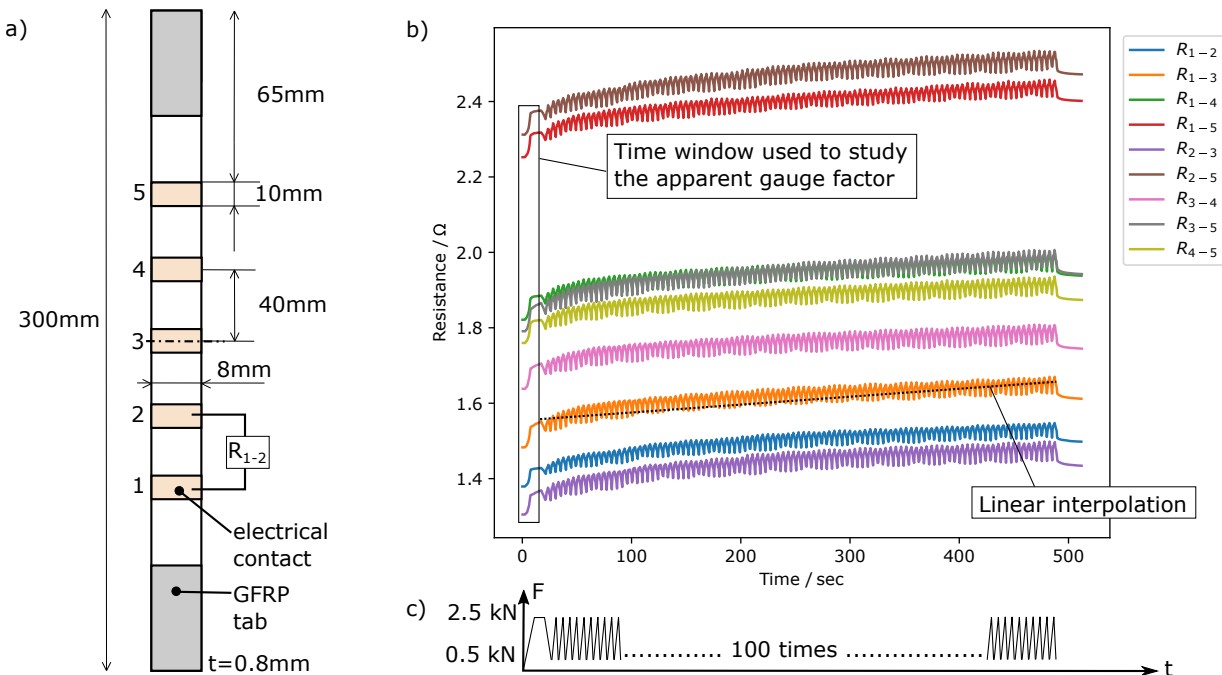

**Figure 5.** Experimental setup and results for measuring the resistance development with strain cycles. (**a**) Specimen dimensions used in this study. (**b**) Results of a sanded specimen contacted with electrodeposited nickel and solder during the first 100 strain cycles. (**c**) Force over time showing the strain cycle of each specimen. * The resistance $R_{2-4}$ was excluded due to a wiring problem that was only found after the experimental study was completed.

## 4. Results

### 4.1. Static Measurements

The contact resistances and coefficients of determination obtained with the methods described in Section 3.4 are displayed in Figure 6. The figure shows large differences between the contact resistances of the specimen that require a logarithmic scale for displaying purposes. Furthermore, the 95 % confidence interval for the contact resistance as obtained from the linear approximation of the sample is displayed.

Overall, electroplated contacts show the smallest contact resistance. Notably, the resistances are in all cases smaller when sanded surfaces are used. Etched surfaces show contact resistances three to four times larger than sanded specimens. Notably, there is no significant difference between the specimen where the cable is connected via solder to the nickel electroplated surface and the specimen where silver epoxy is used to connect cable and electroplated surface.

From the materials analyzed here, both silver paints show the second smallest contact resistance when they are combined with sanded surfaces. Again the contact resistance is about four times larger when the surface is treated with sulphuric acid instead of sanding the surface.

The contact resistances of carbon paint and silver epoxy specimen are one to two orders of magnitude larger than all other materials. Notably, the contact resistance obtained

for silver epoxy that is directly painted onto a carbon fiber surface is 20 times larger than when it is used to connect a cable to an electroplated surface. Carbon paint shows very large measured resistances and does not result in repeatable electrical contact resistances.

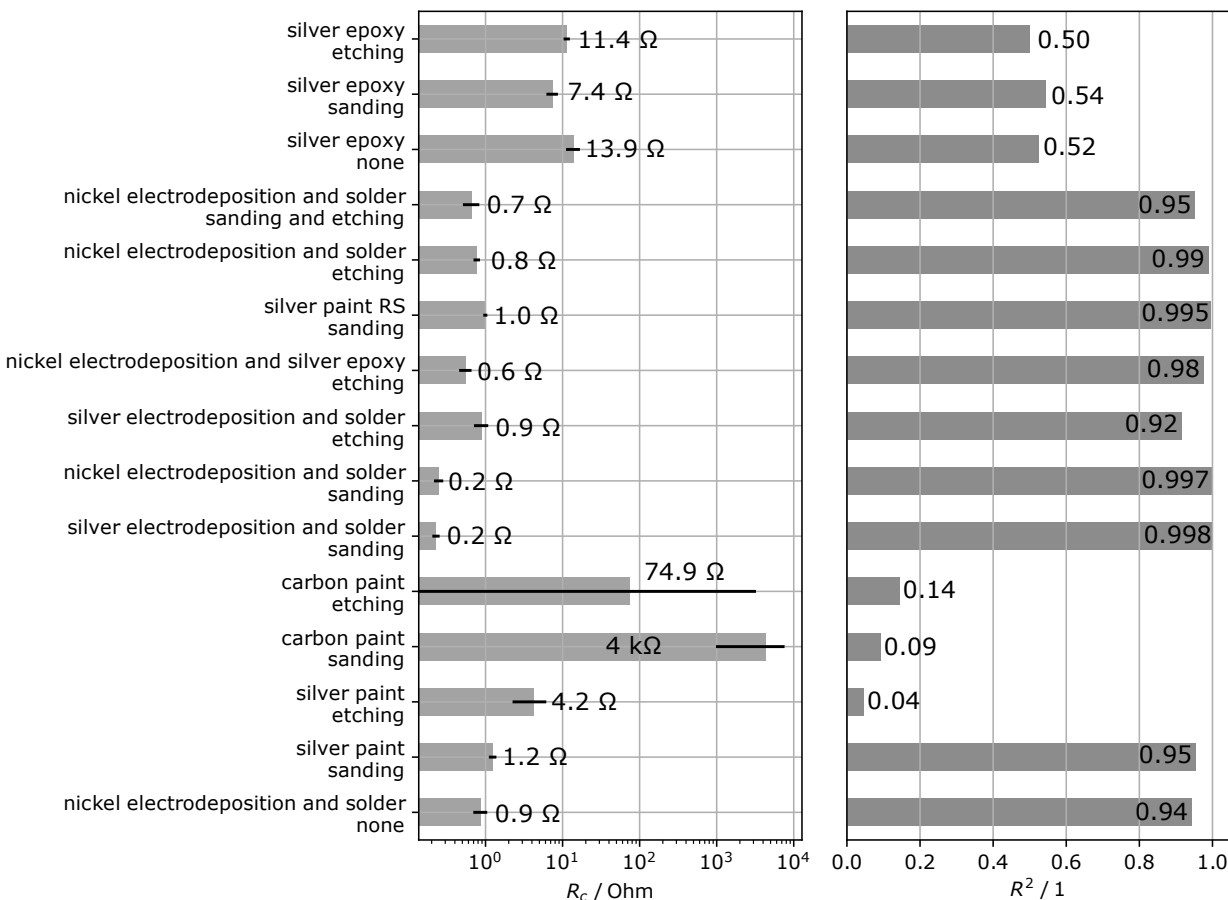

**Figure 6.** Contact resistances and $R^2$ for least square fits of the 2-wire resistance over the distance between contacts. The methodology from Todoroki et al. [7] is used to acquire the results.

The coefficients of determination lead to similar observations, namely that the contacts manufactured with electrodeposition or silver paint show good linearity with $R^2 > 0.95$ and both silver epoxy and carbon cement show small linearity. The largest coefficient of determination is again observed for the specimen contacted with surface sanding and electrodeposited contacts with $R^2 > 0.997$.

### 4.2. Dynamic Experiments

Figure 5 shows a typical result of the resistance development observed in the dynamic experiments during 100 cycles. The results for this contacting method show a relatively linear irreversible increase of the resistance with every strain cycle. We can quantify this slow increase in resistance by calculating the slope of a linear polynomial fit to the data points. The mean linear increase per cycle calculated this way for all specimen is displayed in Figure 7.

Large differences between the contacting materials and methods can be observed. Silver epoxy shows the largest linear increase where the measured resistance increases as much as 6 mΩ per strain cycle. The observed irreversible resistance increase of specimen contacted by electrodeposition is dependent on the surface preparation technique. The irreversible resistance change is overall the smallest when the surface is etched. When the

surface is sanded, a much larger irreversible increase of the resistance is observed. Notably, the absolute resistance increase is larger for sanded surfaces than it is for untreated surfaces.

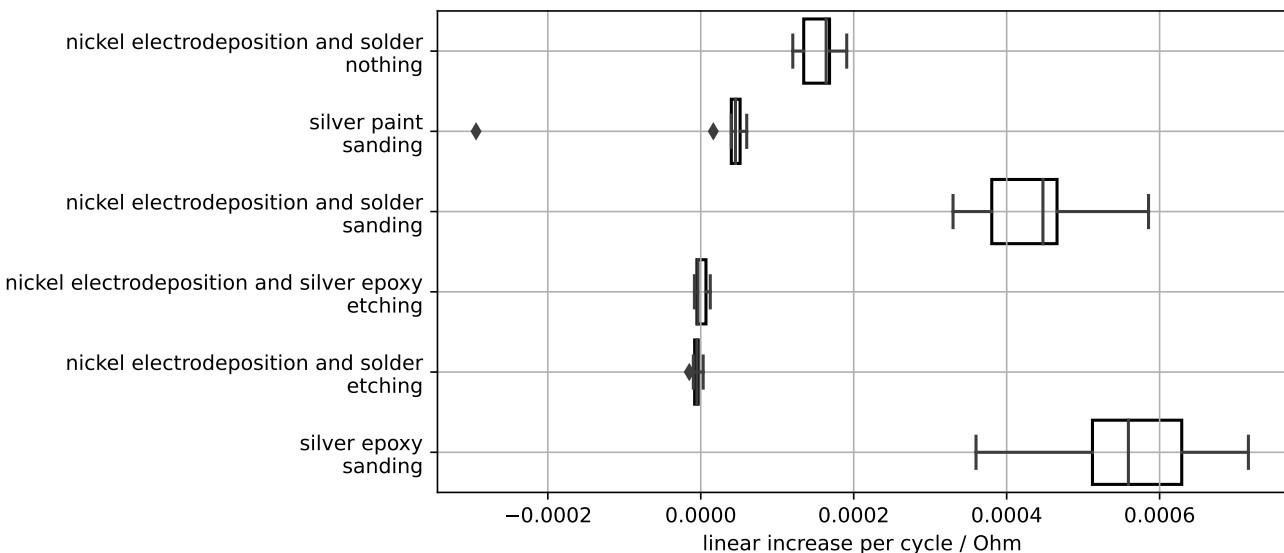

**Figure 7.** Linear irreversible increase of the resistance per cycle.

Next to the analysis of the irreversible resistance change over the cycles, the results displayed in Figure 5 also allows us to calculate the reversible resistance change due to strain. This reversible resistance change can be correlated with the mechanical strain to calculate the apparent gauge factor of the experimental setup. Figure 8 shows the calculated apparent gauge factor for all specimen analyzed in this section. The values are calculated using the initial resistance increase at the beginning of the experiment. Thus, nonlinearities or changes during the strain cycles are not analyzed in this simplified overview. The apparent gauge factor is different between contacting methods. Overall, three groups can be differentiated from the results. Etched surfaces and electrodeposited contacts show the smallest apparent gauge factors. Silver epoxy used on a sanded surface on the other hand shows by far the largest apparent gauge factor. The remaining three variants lie between both of these methods.

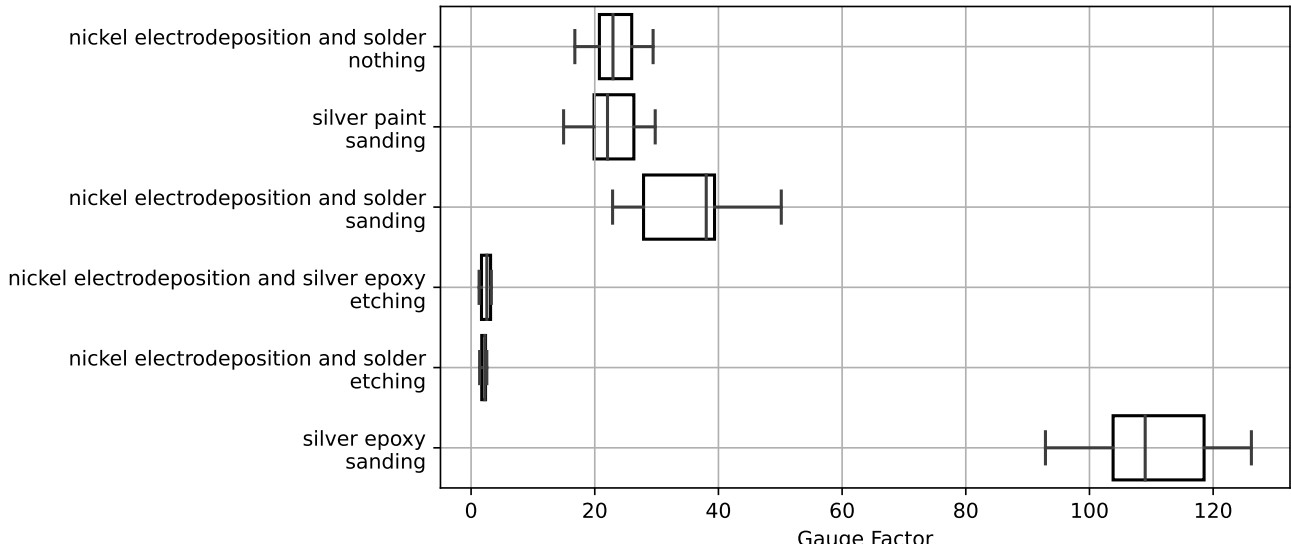

**Figure 8.** Apparent gauge factor for 2-wire measurements with different contacting methods and materials.

## 5. Discussion

The results obtained in both static and dynamic experiments clearly show that both surface preparation and contacting material have a significant impact on the electrical contact to CFRP. There are several repeatable results that can be discussed based on the observations made.

For painted contacts, silver paint has a much smaller contact resistance than silver epoxy. The microscopic evaluation of the contact surfaces displayed in Figures 9 and 10 shows a possible reason for this discrepancy. Silver paint shows the ability to wet a very complex surface morphology and distributes silver particles into very small scratches on the surface. The silver epoxy on the other hand shows this to a lesser extent, which reduces the contact area between silver epoxy and carbon fibers and thereby increases the contact resistance. A possible reason for this discrepancy is the much higher viscosity of the silver epoxy in comparison to silver paint which leads to a decreased contact area between the conductive epoxy and the fibers. This hypothesis is supported by the low contact resistance found for electrodeposited contacts where the cable is attached using the same silver epoxy. This observation shows that the large contact resistance of directly applied silver epoxy stems from the contact between carbon fibers and the silver epoxy and not from some inherent property of the silver epoxy. This hypothesis could also explain the comparably large irreversible resistance increase for silver epoxy specimens observed in dynamic experiments. It seems plausible that a small interface area in which the glue does not penetrate into all small scratches is more susceptible to delaminations that could irreversibly increase the measured resistance.

All electrodeposited specimen show repeatable and small contact resistances with $R^2 > 0.95$. This result is reasonable in comparison to the painted contacts as the electroplating process deposits a layer of highly conductive metal on the surface of CFRP. Figure 11 shows three typical examples of nickel layers deposited onto pultruded CFRP strips. Figure 11a shows a well bonded nickel layer on an etched surface. Figure 11b shows a delaminated nickel layer. An important observation in this image is that the nickel layer shows very well defined circular edges, that comply with the shape of the surface fibers. It is therefore likely, that the nickel originally deposited onto the fiber surface and afterwards delaminated due to some mechanical load. This delamination was likely caused during the handling, cutting, embedding or grinding of the specimen. Notably, an electrodeposited specimen that is furthermore contacted with a standard soldering process shows very similar cross section images with a good bonding (a) and some delaminated parts (b). We therefore cannot report the same solder induced delaminations reported in [7], which might be due to the fact that we use nickel instead of copper for electrodeposition. Figure 11c shows an electroplating specimen where the surface was neither polished nor etched with sulphuric acid. In contrast to the delaminated layer, the nickel layer does not have the same circular shape, which indicates that some boundary layer made from epoxy or sizing exists on the surface that was not removed.

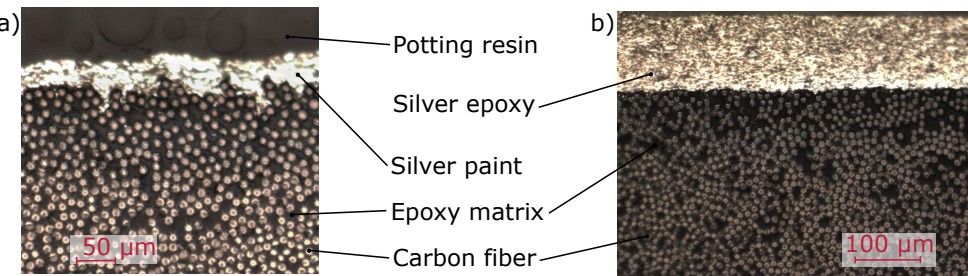

**Figure 9.** Comparison of a cross section contacted with (**a**) silver paint and (**b**) silver epoxy.

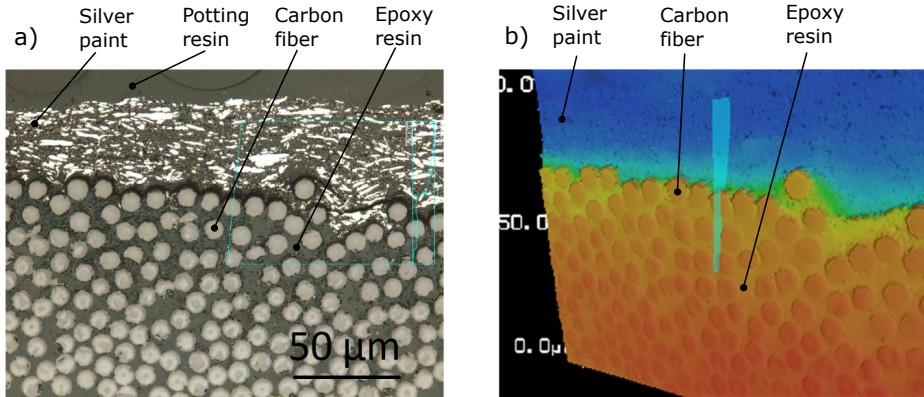

**Figure 10.** Microscopy study of an electrical contact manufactured with silver paint showing (**a**) a microscopic image and (**b**) an image from a laser scanning microscope showing the height profile.

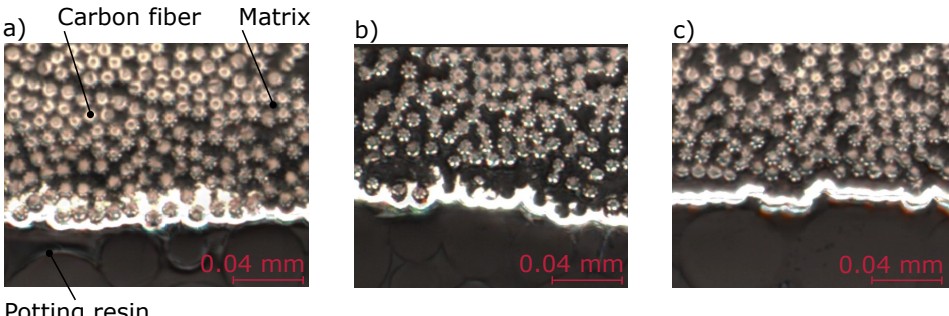

**Figure 11.** Cross section pictures showing (**a**) a well bonded electrodeposition of nickel after etch treatment, (**b**) a delaminated layer of nickel after etch treatment which was likely caused during specimen cutting, (**c**) a poorly deposited layer of nickel due to insufficient surface cleaning.

For these electrodeposited specimens, sanded surfaces show smaller contact resistances when compared to etched surfaces. Figure 12 shows microscopic images of a sanded and an etched surface with the most obvious difference observed between these groups. In some cases, the etching process removes carbon fibers from the matrix. This phenomenon does not occur in the case of the sanding process because sanding abrades the fibers instead of removing only the matrix. This, however, does not seem to be a directly plausible explanation for the larger contact resistance of etched surfaces. It is possible that electrode delaminations occur more frequently for etched surfaces because of loose fibers that are not embedded into the polymer. It is also possible that the loose fibers inhibit the ion transport in the electrolyte to the underlying surface fibers which reduce the metal deposited to the direct surface of the part. Both explanations would reduce the electrical contact area to the direct surface and thereby increase the contact resistance. These hypotheses are however difficult to evaluate based on cross section images, because they inherently only allow evaluating single cross sections and not the full interface area. Furthermore, as displayed in Figure 11b), the preparation process itself is likely to be the cause of at least part of the delaminations observed. In future work, it would be possible to further evaluate this by using Micro-CT scans that analyze the interface area without needing to cut the material or by improving the cutting and embedding process. Improving this could also help to find optimal parameters for the electrodeposition of contacts to CFRP.

Silver paint used on etched surfaces also has a large contact resistance and a very small coefficient of determination when compared to sanded surfaces. We could hypothesize that this is also because surface etching removes some fibers from the epoxy matrix. When silver paint is placed on top, it is possible that it does not penetrate these "loose" fibers everywhere and therefore does not touch the solid surface of the part. It is possible that this problem can be overcome by reducing the etching time to a level where the surface

is sufficiently cleaned but fibers are still fully embedded on the surface. In our view, this could be an interesting analysis to perform because it might lead to contacting technique that has similar properties as electroplated surfaces but is easier to manufacture. Based on the microscopic images obtained in this study, we do not see a clear reason why surface sanding results in smaller contact resistances than surface etching. Further experimental research is necessary to answer this question.

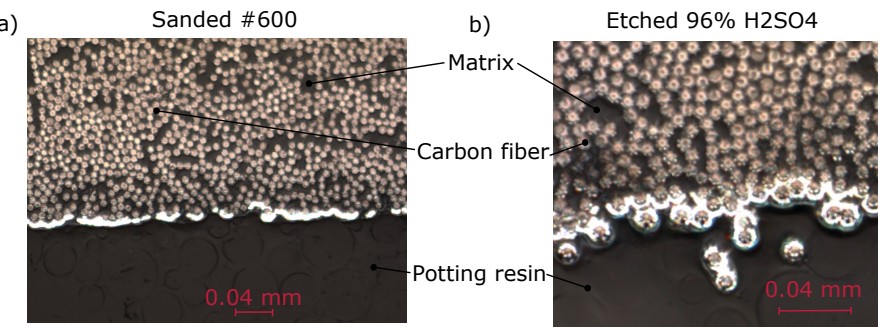

**Figure 12.** Microscopy study of (**a**) sanded and (**b**) etched surfaces contacted with electrodeposition.

In Self-Strain-Sensing applications, we aim to measure the resistance change of the structural part itself. Ideally, an irreversible resistance change should only be observed as a consequence of some failure mechanism in the structural part itself and due to some mechanism caused by the electrical contact. Figure 7 shows that the measured resistance increases irreversibly for most contact pairs. There could be different reasons for this:

- It is possible that mechanical strain delaminates the electrical contact from the specimen surface on a microscopic scale, thereby slowly increasing the electrical resistance.
- Another possible explanation could be a change within the contacting material itself. It seems plausible that strain can irreversibly change the resistivity of filled adhesives because of their percolation based conduction process.
- Yet another possible explanation could be fiber damage occurring due to the sanding process that could cause a successive failure in the part, thereby increasing the measured resistance. This explanation is consistent with the results reported in [10].

Since all sanded specimens show a significantly larger increase of resistance per strain cycle than the etched specimen, it is likely that part of the resistance increase is caused by fiber rupture. However, since the resistance increase of all sanded specimens is different from one another, it seems likely that either delamination or changes in the contact material also play a role in the irreversible resistance change. Surface etching does not cut any carbon fibers and therefore does not show fiber rupture. It also seems plausible that etched and electrodeposited contacts are less susceptible to contact delamination due to strain. Since the etching process washes out part of the surface fibers, the electrodeposition process can fully encapsulate carbon fibers, which increases the interface area between metal and fibers and might improve the mechanical connection.

The observed gauge factor for Self-Strain-Sensing applications should ideally be equal to the intrinsic gauge factor of the CFRP part and independent of the contacting method. The experimental results reported in this article however do not show this ideal behaviour. In contrast, large differences between the apparent gauge factors can be observed. It is very unlikely that this is due to intrinsic differences between individual rods. It is more likely that this difference is due to the contacting method, especially when the very large contact resistances observed in some cases are kept in mind. Etched surfaces contacted with the electrodeposition show the smallest apparent gauge factors between 2 and 3. As shown in [4], typical high strength carbon fibers and rovings such as those used in the pultruded rods of this study are expected to have gauge factors between 1 and 2. Apparent gauge factors significantly larger than this, therefore, indicate that there is a reversible resistance increase of the contact resistance due to strain. In this reasoning, the results obtained for

etched and electroplated surfaces show the most promising results for Self-Strain-Sensing applications based on the dynamic evaluation performed in this section.

## 6. Conclusions

Four main conclusions can be drawn from the results obtained in this article:

1.  Both surface preparation and contact manufacturing technique have significant influence on the quality of electrical contact evaluated based on the requirements for electrical contacts developed in Section 2
2.  From the methods analyzed in this article, the electrodeposition process shows the most promising results in minimizing the electrical contact resistance.
3.  From the methods analyzed in this article, surface etching in combination with an electrodeposition technique shows the most promising results in minimizing the influence of strain on the contact resistance
4.  Electrical contacts manufactured with silver paint have a larger contact resistance than electroplated contacts. Due to their much simpler manufacturing procedure, they are however of high practical interest and could be used in application where the mechanical strain on contacts is limited.

All other combinations analyzed in this article show a significant increase in measured resistance both reversibly and irreversibly and therefore cannot be directly recommended for Self-Strain-Sensing applications. The results obtained in this study are, however, only applicable to the specific material combinations used here and are not expected to be directly transferable to all other cases. Other silver epoxies, silver paints or carbon paints might behave differently than those analyzed in this article. Furthermore, CFRP rods that are manufactured by prepreg or infusion technologies will have different surface conditions than the rods used in this study. This will change the intensity of the surface preparation process necessary to sufficiently clean the CFRP surface. Based on the large differences between different contacting methods observed in this study, we would argue that it would be beneficial for every Self-Strain-Sensing experiment to discuss in detail the surface preparation and contacting technique used to obtain the results.

Furthermore, we see the necessity to further analyze the influence of contact resistance and strain induced changes on four-wire measurements both experimentally and numerically. While the very simple discussions we present in Section 2 allow to conclude the general necessity to analyze contact resistance, they do not allow to quantify the exact influence on the measured electrical potential. It should be feasible to analyze this using a parametric finite element model. The experimental setup we propose in [19] might be a valuable tool in verifying these finite element models by quantifying the electrical surface potential distribution in practical applications.

Eletroplated and etched surfaces show both comparably small contact resistances and a good stability with mechanical strain. They are therefore recommended for Self-Strain-Sensing applications based on the results obtained in this study. However, the 200 strain cycles used in this study do not allow the evaluation of the long-term stability to a sufficient extent. Experiments with significantly more strain cycles are necessary to evaluate this point further.

## 7. Summary and Outlook

In this experimental study, various contacting materials and surface preparation techniques that can be used to electrically contact CFRP are analyzed based on their static performance and their performance under mechanical strain. Overall, the electrodeposition process combined with etched surfaces shows the most promising results for Self-Strain-Sensing applications. This process allows the manufacturing of electrical contacts with low contact resistance and a stable connection under mechanical strain. Silver paint shows a comparably low contact resistance but appears to be more dependent on the exact surface condition and more susceptible to change in the presence of mechanical strain. Both the silver epoxy and the carbon paint used in this study do not show promising results

for manufacturing electrical contacts to CFRP due to their high contact resistance. The results obtained here are, however, only applicable to the specific materials used in this study. Other CFRP manufacturing processes result in different surface properties and might therefore require different surface preparations. The results nevertheless show that contacting methods have different properties and have to be evaluated in detail for a given experimental setup.

A big challenge in electrodeposited contacts appears to be the strength of the mechanical connection between metal and fiber. In our microscopy studies, we frequently find delaminated metal layers (see Figure 11b) that likely occurred during specimen cutting and handling. A possible solution to this could be to change the manufacturing order. Instead of depositing metal onto a cured CFRP sheet, we propose to first deposit metal onto a carbon fiber roving and afterwards use this roving in a pultrusion process. This increases the interface area between metal and carbon fiber significantly and would result in improved mechanical performance. Continuously metalized carbon fibers are state-of-the-art and readily available. However, continuously depositing metal onto the fibers changes their electrical properties and significantly weakens the resulting fiber reinforced plastic. To overcome these drawbacks, we are currently developing and testing an automated electrodeposition procedure that allows to locally deposit metal onto a carbon fiber roving. Figure 13a shows the first results of this research, specifically an experiment in which a thin nickel layer is deposited onto all carbon fibers before embedding into a polymer matrix. Figure 13b shows a macroscopic image in which a 2 cm long section of a roving is plated with nickel. Rovings manufactured in this manner can be connected to measuring equipment with standard soldering processes and used as smart reinforcements in various applications. When used in a pultrusion process, the electroplated section could be uncovered from epoxy using sulphuric acid and connected with a soldering process in a similar manner as described in this article. In other applications, it is possible to perform a defined soldering step before impregnation with epoxy resin. While this approach may bring new challenges such as reduced fiber matrix adhesion, we believe that it could lead to an electrical contact to CFRP with both small contact resistance and stable connection when the material is strained.

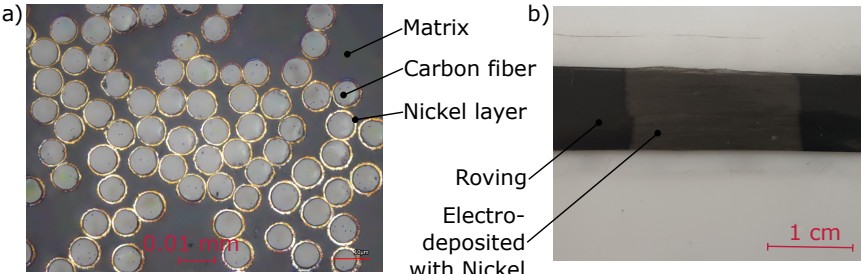

**Figure 13.** Future development of locally electroplated carbon fiber rovings for the automated pultrusion of Self-Strain-Sensing carbon fiber rods with (**a**) showing a typical cross section of an electroplated specimen and (**b**) showing a macroscopic picture of the localized electroplating.

**Author Contributions:** Conceptualization, P.S.; methodology, P.S., S.R.; software, P.S.; validation, P.S.; formal analysis, P.S., S.R.; investigation, P.S., S.R.; resources, M.S.; data curation, P.S.; writing—original draft preparation, P.S.; writing—review and editing, P.S., S.R., M.S.; visualization, P.S.; supervision, M.S.; project administration, P.S., M.S.; funding acquisition, P.S., M.S. All authors have read and agreed to the published version of the manuscript.

**Funding:** This research was funded by German Research Foundation (DFG, Deutsche Forschungsgemeinschaft) Grant No. 447112612. We gratefully acknowledge support by the German Research Foundation and the Open Access Publication Funds of Technische Universität Braunschweig.

**Institutional Review Board Statement:** Not applicable.

**Informed Consent Statement:** Not applicable.

**Data Availability Statement:** The data that support the findings of this study are openly available in Zenodo at https://doi.org/10.5281/zenodo.5534208 (accessed on 22 November 2021).

**Conflicts of Interest:** The authors declare no conflict of interest.

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
