# Peer review of "Comparison of Electrical Contacting Techniques to Carbon Fiber Reinforced Plastics for Self-Strain-Sensing Applications"

_carbon, 2021_

Round 1

Reviewer 1 Report

The paper focuses on the measurement of the electrical resistance of carbon fibre reinforced plastics (CFRPs) for self-sensing applications. A simplified model of a two-fiber sensor is proposed, in which it is demonstrated that the contact resistance is not fully compensated even for four-wire measurements. Based on the analysis using this model, three key requirements on the electrical contacts to CFRPs are proposed: to maximize the current homogeneity, to minimize the contact resistance, and to minimize the impact of the strain on the distribution of the contact resistance. The paper is focused on the second and third requirements, while the first requirement is a subject of a paper submitted elsewhere. The contact resistances are analysed for various contacting techniques, and then the CFRPs are repeatedly stressed to assess the impact of strain on the contact resistance. Based on this analysis the most promising contacting techniques for CFRPs are proposed.   

The paper is of interest to the community dealing with self-sensing applications of CFRPs.

The paper is carefully written; however, there are several sentences that are difficult to understand. Correct them and also correct small grammatical errors (therefor --> therefore, extend --> extent, etc.). I appreciate the outlook section. I recommend the paper for publication in the Journal of Carbon Research once the following recommendations are incorporated into the manuscript.

Define the meaning of y-axis symbols in Figure 1.

What are the dimensions/diameters of the fibres?

Describe the electrodeposition of the contacts in more detail. How is the selective-area deposition of the contacts achieved?

Can you exclude modification of the measured samples when preparing cross-sectional samples for microscopy?

Reviewer 2 Report

In this manuscript, the authors made a comparison on the frequently used electrical contact methods and materials by analyzing their contact resistance to a pultruded carbon fiber reinforced plastic (CFRP) rod. The results show that the contact resistance is highly dependent on both the material used for contacting the laminate as well as the surface preparation technique, which appear to have some practical significance, however, in its present form, it is difficult to recommend the work for publication. Here are some detailed comments:

  1. This paper is not well structured and it is difficult to follow.
  2. The content of manuscript is illogical and less innovation.
  3. The reasons of "surface sanding results in smaller contact resistances when compared to chemical etching" should be fully discussed.
  4. It is difficult to understand for the diagrammatic sketch in Figure 10.

Reviewer 3 Report

  1. There are many grammar mistakes in the main text. The logic always confused for me. For instance, where is experimental part/design/discussion? What do authors want to show?
  2. All techniques were well built in the last few decades, what is new/different of this work comparing with similar work.
  3. The format of three-line table cannot meet the requirement of Journal.
  4. For the figure 6, two photos should be marked as a) or b), then explain it in the caption separately. Figure 7 has a similar issue.
  5. All figures should be re-organized, and make it clearer and orderly.
  6. So many conclusions in the main text, what is the real one?
  7. The writing style is not the same as a regular research paper. I really do not know what authors want to say. Please re-polish or organize the manuscript.

Round 2

Reviewer 2 Report

The manuscript seems to be improved version. Hence, I can state "no comments" and it is recommended for acceptance, if other reviewers and editor agree.

Reviewer 3 Report

Authors have revised all points as comments mentioned. This work should have some benefits for this research field. So, it should be accepted.